# Comparison of the Diagnostic Accuracy of Three Real-Time PCR Assays for the Detection of *Arcobacter butzleri* in Human Stool Samples Targeting Different Genes in a Test Comparison without a Reference Standard

**DOI:** 10.3390/microorganisms11051313

**Published:** 2023-05-17

**Authors:** Ramona Binder, Andreas Hahn, Kirsten Alexandra Eberhardt, Ralf Matthias Hagen, Holger Rohde, Ulrike Loderstädt, Torsten Feldt, Fred Stephen Sarfo, Veronica Di Cristanziano, Sascha Kahlfuss, Hagen Frickmann, Andreas Erich Zautner

**Affiliations:** 1Laboratory Department, Bundeswehr Hospital Hamburg, 20359 Hamburg, Germany; ramonabinder@bundeswehr.org; 2Institute for Medical Microbiology, Virology and Hygiene, University Medicine Rostock, 18057 Rostock, Germany; andreas.hahn@uni-rostock.de (A.H.);; 3Department of Tropical Medicine, Bernhard Nocht Institute for Tropical Medicine & I. Department of Medicine, University Medical Center Hamburg-Eppendorf, 20359 Hamburg, Germany; k.eberhardt@bnitm.de; 4Division of Hygiene and Infectious Diseases, Institute of Hygiene and Environment, 20539 Hamburg, Germany; 5Department of Microbiology and Hospital Hygiene, Bundeswehr Central Hospital Koblenz, 56070 Koblenz, Germany; ralfmatthiashagen@bundeswehr.org; 6Institute of Medical Microbiology, Virology and Hygiene, University Medical Center Hamburg-Eppendorf (UKE), 20251 Hamburg, Germany; rohde@uke.de; 7Department of Hospital Hygiene & Infectious Diseases, University Medicine Göttingen, 37075 Göttingen, Germany; ulrike.loderstaedt1@med.uni-goettingen.de; 8Department of Gastroenterology, Hepatology and Infectious Diseases, University Medical Center Düsseldorf, 40225 Düsseldorf, Germany; torsten.feldt@med.uni-duesseldorf.de; 9Kwame Nkrumah University of Science and Technology, Kumasi 00233, Ghana; stephensarfo78@gmail.com; 10Department of Medicine, Komfo Anokye Teaching Hospital, Kumasi 00233, Ghana; 11Institute of Virology, Faculty of Medicine, University Hospital of Cologne, University of Cologne, 50935 Cologne, Germany; veronica.di-cristanziano@uk-koeln.de; 12Institute of Medical Microbiology and Hospital Hygiene, Medical Faculty, Otto-von-Guericke-University Magdeburg, 39120 Magdeburg, Germany; sascha.kahlfuss@med.ovgu.de; 13CHaMP—Center for Health and Medical Prevention, Otto-von-Guericke-University Magdeburg, 39120 Magdeburg, Germany; 14Institute of Molecular and Clinical Immunology, Medical Faculty, Otto-von-Guericke University Magdeburg, 39104 Magdeburg, Germany; 15Health Campus Immunology, Infectiology and Inflammation (GCI), Medical Faculty, Otto-von-Guericke University Magdeburg, 39104 Magdeburg, Germany; 16Department of Microbiology and Hospital Hygiene, Bundeswehr Hospital Hamburg, 20359 Hamburg, Germany

**Keywords:** *Arcobacter*, *Aliarcobacter*, real-time PCR, evaluation, test comparison, molecular diagnostics

## Abstract

Potential etiological relevance for gastroenteric disorders including diarrhea has been assigned to *Arcobacter butzleri*. However, standard routine diagnostic algorithms for stool samples of patients with diarrhea are rarely adapted to the detection of this pathogen and so, *A. butzleri* is likely to go undetected unless it is specifically addressed, e.g., by applying pathogen-specific molecular diagnostic approaches. In the study presented here, we compared three real-time PCR assays targeting the genes *hsp60*, *rpoB/C* (both hybridization probe assays) and *gyrA* (fluorescence resonance energy transfer assay) of *A. butzleri* in a test comparison without a reference standard using a stool sample collection with a high pretest probability from the Ghanaian endemicity setting. Latent class analysis was applied with the PCR results obtained with a collection of 1495 stool samples showing no signs of PCR inhibition to assess the real-time PCR assays’ diagnostic accuracy. Calculated sensitivity and specificity were 93.0% and 96.9% for the *hsp60*-PCR, 100% and 98.2% for the *rpoB/C*-PCR, as well as 12.7% and 99.8% for the *gyrA*-PCR, respectively. The calculated *A. butzleri* prevalence within the assessed Ghanaian population was 14.7%. As indicated by test results obtained with high-titer spiked samples, cross-reactions of the *hsp60*-assay and *rpoB/C*-assay with phylogenetically related species such as *A. cryaerophilus* can occur but are less likely with phylogenetically more distant species like, e.g., *A. lanthieri*. In conclusion, the *rpoB/C*-assay showed the most promising performance characteristics as the only assay with sensitivity >95%, albeit associated with a broad 95%-confidence interval. In addition, this assay showed still-acceptable specificity of >98% in spite of the known cross-reactivity with phylogenetically closely related species such as *A. cryaerophilus*. If higher certainty is desired, the *gyrA*-assay with specificity close to 100% can be applied for confirmation testing with samples showing positive *rpoB/C*-PCR results. However, in case of a negative result in the *gyrA*-assay, this cannot reliably exclude the detection of *A. butzleri* in the *rpoB/C*-assay due to the *gyrA*-assay’s very low sensitivity.

## 1. Introduction

*Arcobacter butzleri* (homotypic synonym “*Aliarcobacter butzleri*”), formerly addressed as *Campylobacter butzleri* and first described in 1991, is a facultatively pathogenic bacterium from the order Campylobacterales and the family Arcobacteraceae [1,2,3,4,5]. Moreover, in 1991, the genus name *Arcobacter* was proposed [6]. *Arcobacter* spp. are phylogenetically closely related to *Campylobacter* spp. [7]. Nevertheless, the phenotypic characteristics of the genera are partly different [8]. In human patients, *A. butzleri* has been associated with partly severe infections comprising diarrhea including travelers’ diarrhea, enteritis, gangrenous appendicitis, peritonitis, endocarditis and bacteremia and accordingly, it has been regarded as an emerging pathogen since 2002 [9,10,11,12,13]. Nosocomial transmission has been reported [14].

Next to *A. butzleri*, the likely importance of *A. cryaerophilus*, *A. skirrowii* and other species for human enteric disease has been suggested, mostly due to epidemiological associations or based on case reports [12,15,16,17,18]. Other species, such as, e.g., *A. lanthieri*, have been primarily associated with livestock [19], although virulence factors with likely relevance for human disease have been found in their genomes as well [20,21,22]. Altogether, the etiological relevance of *A. butzleri* is considered to be the best established [15,16]. Cell invasion, induction of immune responses and toxin production have been associated with its pathogenic potential [16,18,23]. As expected, due to phylogenetic relatedness, various virulence genes are homologous to those in *Campylobacter* spp. [24]. In particular, invasive and adhesive properties have been demonstrated [25]. Selection of the microorganism under antibiotic pressure can be facilitated by its pronounced resistance to antimicrobial drugs or even multidrug-resistance [16,26]. In a recent meta-analysis on *Arcobacter* spp. [27], high minimum inhibitory concentrations, suggestive of a lack of antimicrobial susceptibility, were recorded, particularly for beta-lactams with up-to 99.2% and 97.4% for penicillin derivates and cephalosporines, respectively, followed by macrolides with up-to 39.8%, fluoroquinolones with up-to 14%, aminoglycosides with up-to 12.9% and tetracyclines with up-to 7.1%. Further, sequencing approaches have shown a high diversity of sequence types, mostly without clear-cut associations to specific hosts or geographic regions [24].

Livestock, including poultry and pigs, has been identified as a reservoir of *A. butzleri*, and raw meat products, as well as contaminated water, are of relevance for the pathogen’s transmission [15,24,28,29,30,31]. Accordingly, *A. butzleri*-associated disease is considered zoonotic [32]. Meat contamination is assumed to be caused by spillage of gastrointestinal fluids from animals during the slaughtering process [12]. Adverse environmental conditions such as food processing and storage can be survived by the microorganism, supporting its spread to human individuals [16].

Culture-based isolation and differentiation of *A. butzleri* is still not a standardized routine procedure in diagnostic laboratories assessing human sample materials [15,32]. The lack of standardized diagnosis has also been blamed for the lack of reports of *A. butzleri*-associated disease in regions where the microorganisms are known to be highly prevalent like, e.g., in Nigeria [33]. Though cultural growth of *Arcobacter* spp. in microaerobic or aerobic atmosphere is feasible, and—other than *Campylobacter* spp.—the microorganisms even grow at low temperature of 15 °C, common growth protocols suggest enrichment steps, i.e., use of selective broths and agars to suppress concomitant bacterial flora, as well as incubation times of 4–5 days [13,23,34]. *A. butzleri* also grows on standard agars such as blood agar, chocolate agar and MacConkey agar under standard conditions such as a temperature of 37 °C and an atmosphere enriched with 5% CO_2_, with colonies showing positive results in cytochrome oxidase and motility testing [13]. Biochemical differentiation is challenging [12,13], and the reliability of matrix-assisted laser-desorption-ionization time-of-flight mass spectrometry (MALDI) largely depends on the quality of the underlying database as reported elsewhere [18]. Accordingly, *A. butzleri*-associated gastroenteritis is assumed to frequently go undetected, making an estimation of the role of this microorganism in infectious gastroenteritis challenging [12]. In line with this, a previously reported association of *Arcobacter* spp.-isolations with 0.11% to 1.25% of diarrhea cases might considerably underestimate the microorganisms’ true prevalence [18].

Considering the challenges of traditional culture-based *Arcobacter* spp.-detection and identification in the diagnostic routine setting [12,13,15,18,23,32,33,34], molecular diagnostic approaches based on PCR or real-time PCR with or without subsequent sequence analysis were established [13,35]. Of note, however, early *Arcobacter* spp.-specific PCR assays targeting genes such as *gyrA* or the 16S and 23S ribosomal RNA genes showed restricted diagnostic accuracy including cross-reactions with non-target species [35]. For various *Arcobacter* spp., including *A. butzleri*, fluorescence resonance energy transfer (FRET)-based and hybridization probe-based real-time PCR assays have been introduced, established and applied for human diagnostic, agricultural and environmental use [17,36,37,38,39,40,41,42,43,44,45,46,47]; however, the validation studies were usually based on quite limited sample counts.

In the study presented here, the aim was to contribute to available evidence on the diagnostic accuracy of selected *A. butzleri*-specific real-time PCR assays [36,37,38] based on a test comparison without a reference standard [48], applying latent class analysis [49] and using a sample collection with unknown *A. butzleri* prevalence but expected high pretest probability due to a known high local abundance of this microorganism [26].

## 2. Materials and Methods

### 2.1. Residual Volumes of Sample Materials Used for the Test Comparison, Inclusion and Exclusion Criteria

The test comparison of the three assessed real-time PCR assays for the detection of *A. butzleri* in human stool samples was conducted in two steps. In a first step, 65 stool samples were spiked to high final concentrations of about 10^7^ colony-forming-units (cfu) per µL with clinical and environmental *A. butzleri* strains (*n* = 30), *A. cryaerophilus* strains (*n* = 22), livestock-associated *A. lanthieri* strains (*n* = 12) [19] and a single *Campylobacter coli* strain as an outstander from a strain collection at the University of Magdeburg, Germany. Those spiked samples with high pathogen density were used to assess the general applicability of the compared assays, applying samples with known expected results in the first but superficial proof-of-principle. Dilution steps were not included in the proof-of-principle spiking experiments; respective dilution series to identify the applied real-time PCRs’ technical detection thresholds were performed with standardized positive control plasmids as described below instead.

In a second step, residual volumes of nucleic acid extraction eluates of stool samples taken from Ghanaian individuals with an expected high pretest probability of being colonized or infected with *A. butzleri* were included in a test comparison without a reference standard. The high pretest probability was assumed due to a known high local abundance of the microorganism in Western Africa [26,33]. Results of other diagnostic tests targeting this parameter, such as, e.g., cultural growth, were not available for these materials. The sample collection comprised a total of 1570 samples collected in the course of both a study on Ghanaian HIV patients [50,51] and an assessment of the epidemiology of gastroenteric pathogens in Ghanian children <2 years of age about 10 years in the past (unpublished).

All samples containing sufficient material for all compared test assays were included in the assessment. Samples showing sample inhibition in the inhibition control PCR as detailed below were subsequently excluded from the calculation of diagnostic accuracy parameters. In line with the ethical requirements for this test assessment as detailed below, complete anonymization of the samples was required for the analysis; thus, patient specific details such as age, sex or clinical symptoms cannot be provided for this study. This is an admitted deviation from the Standards of Reporting Diagnostic Accuracy (STARD) criteria [52].

### 2.2. Nucleic Acid Extraction and Storage

Nucleic acids had been extracted, applying the QIAamp stool DNA mini kit (Qiagen, Hilden, Germany) as described by the manufacturer and stored at −80 °C until application for the test comparison.

### 2.3. Real-Time PCR Assays Comparatively Applied for the Detection of Arcobacter butzleri

The compared assays comprised two hybridization probe-based real-time PCRs targeting the *rpoB/C* gene [37] and the *hsp60* gene [38] of *A. butzleri* as well as a FRET-based real-time PCR designed to target the *gyrA* gene of *A. butzleri*, *A. cryaerophilus*, *Arcobacter cibarius* and *Arcobacter nitrofigilis* at different distinguishable melting temperatures [36]. The protocols were adapted to be run on Corbett Q cyclers (Qiagen, Hilden, Germany), resulting in minor modifications from the originally published run conditions [36,37,38]. Details on the applied oligonucleotides, reaction chemistry and run protocols are provided in Appendix A, Table A1, Table A2 and Table A3. A PCR-grade water-based negative control and a plasmid-based positive control were included in each run. The sequence insert of the positive control plasmid, which had been inserted in a pEX A128 vector backbone (eurofins Genomics, Luxemburg), is shown in Appendix A, Table A3. Based on dilution series of the positive control plasmid, technical detection thresholds of 37.1 copies per µL for the *hsp60* and the *gyrA* PCR as well as of 370.5 copies per µL for the *rpoB/C* PCR were estimated, applying the internet-based software “Calculator for determining the number of copies of a template” (URI Genomics and Sequencing Center, https://cels.uri.edu/gsc/cndna.html, last accessed on 8 March 2023). Sample inhibition was confirmed or ruled out, applying a previously described real-time PCR targeting a Phocid herpes virus (PhHV) sequence as described elsewhere [53].

### 2.4. Diagnostic Accuracy Estimation, Agreement and Comparison of Obtained Cycle Threshold (Ct) Values

In the first step of the evaluation, the spiked stool samples were used to assess the general applicability of the compared real-time PCR assays. Accordingly, obtained cycle threshold (Ct) values were just descriptively recorded and assessed. Next to this, this step was conducted to identify the *A. butzleri*- and *A. cryaerophilus*-specific melting temperature for the adapted run conditions of the *gyrA*-PCR, in addition to the use of the positive control plasmid containing an *A. butzleri*-specific *gyrA*-sequence for this purpose.

In the second step, latent class analysis (LCA) [48,49,54] was applied to estimate the diagnostic accuracy—i.e., sensitivity and specificity for the detection of *A. butzerli*—of the compared real-time PCR assays. In short, LCA is a variant of structural equation models. Thereby, a latent non-observable variable, i.e., the “true” infection or colonization status of the tested individuals with *A. butzleri*, is estimated over directly observed variables, i.e., the recorded test results for different *A. butzleri*-specific target sequences [48,49,54]. Thereby, only real-time PCR signals in the *gyrA*-PCR with *A. butzleri*-specific melting temperature were counted as *A. butzleri* positive in this assay, while *gyrA*-PCR signals with melting temperatures suggestive of *A. cryaerophilus* or other microorganisms were considered as *gyrA*-negative for the target pathogen *A. butzleri*. As the *gyrA* sequence is highly under mutation pressure both naturally and under quinolone treatment, a melting temperature deviation from the expected melting temperature within the ±1 °C range was still accepted as species specific. In addition, LCA was applied to perform a diagnostic accuracy-adjusted prevalence estimation for *A. butzleri* in the assessed sample population. In line with the suggestions by Landis and Koch [55], Fleiss’ kappa was calculated for the agreement between the compared real-time PCR assays and interpreted as described in [55]. In addition, measured cycle threshold (Ct) values were recorded and descriptively compared. Typical sigmoid-shaped real-time amplification curves were considered as positive signals without specific Ct value cut-offs for this assessment. All statistical assessments were conducted with the software Stata/IC 15.1 for Mac 64-bit Intel (College Station, TX, USA).

### 2.5. Ethics

Ethical clearance for the anonymized use of residual volumes of sample materials for test comparison purposes was provided by the medical association of Hamburg, Germany, (reference number: WF-011/19, obtained on 11 March 2019) without further requirement for informed consent.

## 3. Results

### 3.1. Quantitative and Qualitative Results Obtained with Spiked Sample Materials

All applied *A. butzleri*-specific real-time PCRs correctly identified all 30 stool samples spiked with *A. butzleri* at high titer as described above. Cross-reactions with high-titer *A. cryaerophilus*-spiked stool samples were observed in 6 out of 22 cases for the *hsp60*-PCR and in 16 out of 22 cases for the *rpoB/C*-assay. For the *hsp60*-PCR, the mean Ct-value difference between *A. butzleri*-spiked samples and *A. cryaerophilus*-spiked samples was 14.9; for the *rpoB/C*-PCR, it was 4.5 Ct steps. The *gyrA*-assay showed an expected reaction with 21 out of 22 *A. cryaerophilus*-spiked samples, albeit with a clearly discriminable melting-temperature compared to *A. butzleri*-spiked samples. Of note, a moderate Ct-value difference of 7.3 was seen for the *gyrA*-PCR in comparison of *A. butzleri*- and *A. cryaerophilus*-spiked reference samples as well. All 12 assessed samples spiked with *A. lanthieri* showed negative results in the three assessed real-time PCR assays. Details are provided in Table 1 and its footnote. The single included “outstander” sample spiked with *C. coli* showed a signal in the *rpoB/C*-PCR only at a very high Ct value of 35.0, i.e., 18.6 Ct steps after the mean value of the samples spiked with *A. butzleri*.

### 3.2. Sensitivity and Specificity of the Assays as Calculated Based on Latent Class Analysis (LCA), Agreement between the Compared Assays and Accuracy-Adjusted Prevalence Estimations

For the LCA-based assessment of diagnostic accuracy, 75 samples showing PCR inhibition were excluded, resulting in a total number of *n* = 1495 included residual volumes of Ghanaian stool sample materials. As detailed in Table 2, the number of positive PCR signals within these included samples ranged from *n* = 30, as recorded with the *gyrA*-PCR, to *n* = 245, as recorded with the *hsp60*-PCR. Focusing on diagnostic accuracy, calculated sensitivity showed a broad spectrum ranging from 12.7% to 100% with *rpoB/C*-PCR, *hsp60*-PCR and *gyrA*-PCR in declining order of sensitivity. Calculated sensitivity >95% was observed for the *rpoB/C*-PCR only, albeit with a broad 95%-confidence interval. The calculated specificity values were much closer together, ranging from 96.8% to 98.8% with *gyrA*-PCR, *rpoB/C*-PCR and *hsp60*-PCR in declining order of specificity. The agreement kappa calculated for the results of all three compared PCRs was only moderate, as shown in Table 2, and mainly driven by the influence of the *gyrA*-PCR results. If only the *hsp60*-PCR and the *rpoB/C*-PCR were compared, the agreement kappa would rise to almost perfect with a value (95%-confidence interval) of 0.812 (0.771–0.852) (Table 2). Concordance and discordance between the individual real-time PCR assays are visualized in Table 3. Based on the LCA-assessments, a prevalence of 14.8% colonization or infection with *A. butzleri* was calculated for the assessed Ghanaian stool sample collection (Table 2).

### 3.3. Comparison of the Recorded Cycle Threshold (Ct) Values

Recorded median cycle threshold values ranged from 32 to 40.5 over all three assessed real-time PCRs for the 1495 samples without recorded PCR inhibition included in the LCA assessment and excluding the above-mentioned *gyrA*-PCR results with melting temperatures non-indicative for *A. butzleri*. Thereby, *hsp60*-PCR, *rpoB/C*-PCR and *gyrA*-PCR were arranged in inclining order of median Ct values (Table 4). While the measured median Ct values of *hsp60*-PCR and *rpoB/C*-PCR were quite similar, the values obtained with the *gyrA*-PCR were on average 6 to 9 Ct steps higher.

## 4. Discussion

Considering the yet-poor standardization of the diagnosis of *Arcobacter butzleri* in human stool samples in the routine diagnostic setting, the study was conducted to comparatively assess the diagnostic accuracy of three published real-time PCR assays with different target genes in a latent class analysis (LCA)-based test comparison without a reference standard using a sample collection with high pretest probability. The assessment led to a number of results.

First, the initial assessment with stool samples spiked at high titers with either *A. butzleri* or the phylogenetically closely related *A. cryaerophilus* indicated that all assessed real-time PCR assays are not perfectly specific for *A. butzleri* but have the potential of cross-reaction with phylogenetically related *Arcobacter* species. The lack of positive real-time PCR signals with samples spiked with 12 *A. lanthieri* strains at least indicates that such cross-reactions do not necessarily occur with all representatives of the genus. The observed cross-reaction of the *rpoB*-assay with the single “outstander” sample, which was high titer-spiked with *Campylobacter coli*, was considered as hardly relevant for the diagnostic situation due to the extraordinarily large associated Ct value shift. While the reaction with *A. cryaerophilus* was expected for the *gyrA*-assay due to its design [36] and could be easily discriminated from reactions with *A. butzleri* due to distinct melting temperatures, such a discrimination was unfeasible for the hybridization probe-based *rpoB/C*-assay and *hsp60*-assay. However, Ct-values shifts of 14.9 when comparing *A. butzleri*-spiked samples and *A. cryaerophilus*-spiked samples in the *hsp60*-PCR, and of 4.5 when comparing them in the *rpoB/C*-PCR, indicated base-mismatching-associated reduced likeliness of cross-reaction with *A. cryaerophilus*. This might result in lower susceptibility of the hybridization probe assays to non-specific reactions in case of use with clinical samples without spiking-associated exorbitantly high concentrations of pathogen DNA. Nevertheless, the results confirm the previously described challenges regarding the design of PCR assays with reliable selectivity for *A. butzleri* [35].

Second, LCA-based prevalence estimation of 14.7% *A. butzleri*-positive cases among the assessed Ghanaian stool samples confirmed the study’s assumption that the chosen specimen collection was associated with high pre-test probability. This finding matches previous reports on *A. butzleri* prevalence rates in Ghanian livestock being even higher than *Campylobacter* spp.-prevalence rates [56] next to reports on high enteric infection and colonization rates with bacterial pathogens in Ghanaian individuals [57,58] and on high *A. butzleri*-prevalence rates in Western African Nigeria [33]. In line with the ethical clearance of the here-presented study, the residual volumes of the samples were completely anonymously assessed; thus, no statements on epidemiological features such as age, sex and symptom associations can be provided. As stated above, this design is an admitted deviation from STARD criteria [52]. However, it is nevertheless in line with diagnostic routine conditions, despite the long storage of extracted DNA, because as-good-as-possible diagnostic accuracy for the detection of the target pathogen is required irrespective of the epidemiological background.

Third, and in line with previous findings [35], the LCA-assessment indicated considerable difference regarding the diagnostic accuracy of the assessed three published real-time PCR assays for the detection of *A. butzleri* [36,37,38]. The FRET-assay targeting the *gyrA* gene [36] showed the best specificity of close to 100%, albeit for the price of a very low sensitivity of less than 15%. Considering the high Ct values measured for the *gyrA*-PCR-positive Ghanian sample materials, it is likely that target DNA quantities close to the PCR’s detection threshold have been the reason for the poor sensitivity result. A 10-fold higher detection threshold observed with the dilution series of the positive control plasmid compared to the *hsp60*-assay and the *rpoB/C*-assay as described above also speaks in favor of this conclusion. The hybridization probe-based *hsp60*-assay showed intermediate sensitivity of only slightly less than 95% but the comparably worst specificity of less than 97%. The latter is also the reason for the calculated sensitivity being lower than the calculated sensitivity of the *rpoB/C*-assay, although more positive results were obtained with the *hsp60*-assay from the sample collection. Focusing on the observed higher likeliness of the *rpoB/C*-assay to cross-react with *A. cryaerophilus*-spiked samples compared to the *hsp60*-assay, the latter’s worse specificity might be considered surprising. However, the melting curve analysis of the non-*A. butzleri*-associated positive results in the *gyrA*-assay indicated a very low prevalence of only six samples positive for *A. cryaerophilus*, of which four showed negative reactions in the *hsp60*-assay. Due to the quantitatively low relevance of *A. cryaerophilus* as a potentially cross-reacting agent in the Ghanaian sample collection, its effect on the diagnostic accuracy estimations has most certainly been negligible. The *rpoB/C*-assay was the only assay with a calculated sensitivity of more than 95%, at least if the uncertainty due to the broad 95%-confidence interval is accepted, associated with a still-acceptable specificity of slightly more than 98%. In spite of the large confidence interval of the *rpoB/C*-PCR’s estimated sensitivity, the true sensitivity of the assay is nevertheless likely to be actually high due to the observed combination of (a) a high absolute number of positive PCR signals; (b) the almost perfect agreement of its results with the results of the *hsp60*-assay, which showed sensitivity only slightly lower than 95%; and (c) the quite-acceptable specificity. The *rpoB/C*-assay’s relatively good specificity may seem surprising considering the high rate of observed matching between positive *rpoB/C*-PCR results and positive *gyrA*-PCR results, with melting curves indicative of non-*A. butzleri* DNA. However, the observed non-*A. butzleri*-specific melting curves may have been due to co-colonization of the stool donators’ gut with both *A. butzleri* and microorganisms with DNA more readily reacting with the *gyrA*-assay and so, it cannot be assumed for certain that the associated *rpoB/C*-PCR results must have been false positive in all these instances. The same applies to positive *hsp60*-PCR results in concordance with non-*A. butzleri*-specific positive *gyrA*-PCR signals. This co-colonization hypothesis is supported by the finding of quite-similar mean Ct values in *hsp60*-PCR and *rpoB/C*-PCR, respectively, irrespective of the observation of *A. butzleri*-specific or non-*A. butzleri*-specific melting temperatures in the *gyrA*-RCR. In contrast, the broad spectrum of observed melting temperatures in the case of non-*A. butzleri*-specific positive *gyrA*-PCR signals indicates that several phylogenetically related non-target microorganisms potentially associated with cross-reactivity were abundant in the sample collection.

This study has a number of limitations. First, a culture-based reference standard for the test comparison was not available. Respective attempts were unfeasible due to the retrospective design of the study. However, due to the lack of standardization regarding the cultural growth-based diagnosis of *A. butzleri*, uncertainties regarding the sensitivity and specificity of such an approach, even from fresh sample materials in the case of a prospective study design, would have limited its value as a reliable reference standard for the test comparison. Second, the choice of microorganisms used for the initial spiking experiments was restricted by the availability within the research group. However, the spiking approach was considered as no more than an initial proof-of-principle, while the main study was based upon the LCA approach. Third, lacking funding of this investigator-initiated investigation made sequencing-based confirmatory testing from all obtained PCR amplicons unfeasible; thus, LCA was applied for the diagnostic accuracy estimation. Due to this lack of a sequencing option, unfortunately, *gyrA* PCR signals with melting temperatures different from the expected values for *A. butzleri* and *A. cryaerophilus* could not be resolved. Fourth, and as repeatedly stated above, the ethical requirement of thorough anonymization for the test comparison made any comparison of *A. butzleri* detections, both qualitatively and with focus on recorded Ct values, with clinical symptoms of the assessed individuals unfeasible. Future studies should be conducted in order to address this highly relevant issue and to estimate the clinical relevance of such real-time PCR findings. Fifth, and as an intrinsic limitation of the chosen mathematic approach, the principle of LCA implies that the calculated diagnostic accuracy does not necessarily refer specifically to *A. butzleri* but to a meta-structure sharing genetic elements detected by the three compared assays. This might as well mean a combination of *A. butzleri* and other phylogenetically closely related microorganisms abundant in the assessed Ghanaian stool samples. The observation of potential cross-reactivity within the *Arcobacter*/“*Aliarcobacter*” genus described in this study makes this option rather likely. Accordingly, LCA can help to estimate prevalence rates on a population level but cannot decide on the correctness of an individual PCR result.

## 5. Conclusions

In conclusion, imperfect diagnostic accuracy was observed for all assessed real-time PCR assays for the detection of *A. butzleri*. In comparison, the *rpoB/C*-assay showed the best performance characteristics, with sensitivity >95%, if residual uncertainty due to an observed broad 95% confidence interval is accepted, and with specificity >98%. Of note, the assay might be less reliable in settings where the discrimination of *A. butzleri* and *A. cryaerophilus* is of importance, e.g., due to relevant prevalence of the latter. For confirmation testing in cases of positive results obtained with the *rpoB/C*-based screening assay, the application of the highly specific *gyrA*-assay may be considered. In the case of a matching positive result with an *A. butzleri*-specific melting temperature, the diagnosis of *A. butzleri* can be considered as confirmed with high reliability. In the case of a negative *gyrA*-PCR result, however, the diagnosis of *A. butzleri* is not excluded due to the *gyrA*-assay’s low sensitivity. Finally, due to the observed high prevalence of *A. butzleri* in Ghanaian stool samples, future studies on its etiological relevance in the Ghanaian population seem advisable because the present study’s exclusive focus on technical aspects does not allow us to answer this.

## Figures and Tables

**Table 1 microorganisms-11-01313-t001:** Positive real-time PCR signals in stool samples spiked with either *A. butzleri*, *A. cryaerophilus* or *A. lanthieri*.

rReal-Time PCR Target	*gyrA*	*rpoB/C*	*hsp60*
Number and percentage of positive signals with samples spiked with *A. butzleri*, *n*/*n* (%)	30/30 (100%)	30/30 (100%)	30/30 (100%)
Ct values measured with samples spiked with *A. butzleri*, mean (±SD)	22.2 (±4.6)	16.4 (±3.8)	16.6 (±3.4)
Melting temperature in °C (±SD) with *A. butzleri*	65.7 (±<0.1)	n.a.	n.a.
Number and percentage of positive signals with samples spiked with *A. cryaerophilus*, *n*/*n* (%)	0/22 (0%) °	16/22 (72.7%)	6/22 (27.3%)
Ct values measured with samples spiked with *A. cryaerophilus*, mean (±SD)	n.a.	20.9 (±2.7)	31.5 (±0.6)
Melting temperature in °C (±SD) with *A. cryaerophilus*	60.5 (±0.3)	n.a.	n.a.
Number and percentage of positive signals with samples spiked with *A. lanthieri*, *n*/*n* (%)	0/12 (0%)	0/12 (0%)	0/12 (0%)
Ct values measured with samples spiked with *A. lanthieri*, mean (±SD)	n.a.	n.a.	n.a.
Melting temperature in °C (±SD) with *A. lanthieri*	n.a.	n.a.	n.a.
Significance level P for differences of the measured Ct values of *A. butzleri* and *A. cryaerophilus* *	n.e.	0.0003	<0.0001

*n* = number; % = percent; Ct = cycle threshold; n.e. = not estimable; n.a. = not applicable; SD = standard deviation. * Calculation based on Mann–Whitney U testing. ° Twenty-one out of twenty-two (95.5%) samples were positive with clearly distinguishable melting temperature (see main text) and a significance (*p* = <0.0001) for higher Ct values of 29.5 (±4.1) compared to the samples spiked with *A. butzleri*.

**Table 2 microorganisms-11-01313-t002:** Agreement kappa between the three compared real-time PCR assays targeting *A. butzleri* as well as sensitivity, specificity and accuracy-adjusted prevalence as calculated with latent class analysis (LCA) based on the assessment of 1495 samples, not showing PCR inhibition with high pre-test probability.

Assay	Total Number (*n*) of Included Samples	Positives (%)	Sensitivity (0.95 CI)	Specificity (0.95 CI)	Kappa (0.95 CI)
*gyrA*	1495	30 (1.91)	0.1267 (0.0876, 0.1797)	0.9984 (0.9936, 0.9996)	0.436 (0.403, 0.472)
*rpoB/C*	1495	244 (16.32)	1 (0, 1)	0.9818 (0.9499, 0.9935)	
*hsp60*	1495	245 (16.39)	0.9298 (0.7513, 0.9831)	0.9688 (0.9576, 0.9771)	
Prevalence (0.95 CI)	0.1477 (0.1258, 0.1726)				

CI = confidence interval.

**Table 3 microorganisms-11-01313-t003:** Cross-table detailing mismatches between the real-time PCR assays targeting *A. butzleri*.

		*gyrA*		*rpoB/C*		*hsp60*	
		Negative	Positive	Negative	Positive	Negative	Positive
*gyrA*	Negative	1465					
	Positive	0	30				
*rpoB/C*	Negative	1249	2	1251			
	Positive	216	28	0	244		
*hsp60*	Negative	1246	4	1212	38	1250	
	Positive	219	26	39	206	0	245

Green = matching results; Red = mismatching results; Black = not filled in to avoid repetition.

**Table 4 microorganisms-11-01313-t004:** Recorded cycle threshold (Ct) values of the real-time PCRs targeting *A. butzleri*.

	*n*	Mean (SD)	Median (q25, q75)
*gyrA*	30	39.83 (2.41)	40.5 (39, 41)
*rpoB/C*	244	33.55 (1.77)	34 (33, 35)
*hsp60*	245	31.86 (1.58)	32 (31, 33)

SD = standard deviation; q25 = 25%-quartile; q75 = 75%-quartile.

## Data Availability

All relevant data are presented in the article. Raw data can be provided on reasonable request.

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
