# Peer review of "Comparison of the Diagnostic Accuracy of Three Real-Time PCR Assays for the Detection of Arcobacter butzleri in Human Stool Samples Targeting Different Genes in a Test Comparison without a Reference Standard"

_microorganisms, 2023, doi:10.3390/microorganisms11051313_

Round 1

Reviewer 1 Report

Review on: Microorganisms Manuscript titled “Comparison of the diagnostic accuracy of three real-time PCR 3 assays for the detection of Arcobacter butzleri in human stool 4 samples targeting different genes in a test comparison without 5 a reference standard” by Binder et al.

This manuscript deals with the usefulness of three RT-PCRs previously published for the detection of A. butzleri. This methodical works uses spiked fecal samples with A. butzleri and closely related organisms and a field study using 10 year old extracted nucleic acid of stool samples from Ghanaian individuals. Ethical compliance was gained. None of these three previously described PCR methods were able to have high specificity and enough sensitivity as stated by the authors for a valuable routine detection method for A. butzleri. Authors discussed the main limitations of the very study.

General comments:

The study is accurate prepared, ethical requirements have been accomplished and main limitations have been identified.

Nevertheless different spiking levels would have been of interest to the study to set a correct CT value. If these results are available please add to the manuscript.

Add CT value used for the field study for all three RT-PCRs.

Specific comments:

Line 177-178 and line 208-211: As the gyrA is highly under mutation pressure (naturally and under quinolone treatment) please state how you assessed possible influence on the melting curve results. Please state here or elsewhere in the manuscript.

Delete line 265-281 as it is a repetition of line 248-264.

Line 341-343: add despite the long storage of extracted DNA

Line 372: delete “r” before “high”

Author Response

Reviewer 1, first comment:

The study is accurate prepared, ethical requirements have been accomplished and main limitations have been identified. Nevertheless different spiking levels would have been of interest to the study to set a correct CT value. If these results are available please add to the manuscript.

Authors:

We have now clarified that dilution steps were not included in the proof-of-principle spiking experiments, respective dilution series to identify the applied real-time PCRs’ technical detection thresholds were performed with standardized positive control plasmids as described below instead (methods chapter “Residual volumes of sample materials used for the test comparison, inclusion and exclusion criteria”, first paragraph, new last sentence).

Reviewer 1, second comment:

Add CT value used for the field study for all three RT-PCRs.

Authors:

As requested, we have clarified that typical sigmoid-shaped real-time amplification curves were considered as positive signals without specific Ct value cut-offs for this assessment (methods subheading “Diagnostic accuracy estimation, agreement and comparison of obtained cycle threshold (Ct) values”, second paragraph, new fifth sentence).

Reviewer 1, third comment:

Line 177-178 and line 208-211: As the gyrA is highly under mutation pressure (naturally and under quinolone treatment) please state how you assessed possible influence on the melting curve results. Please state here or elsewhere in the manuscript.

Authors:

We have specified that as the gyrA sequence is highly under mutation pressure both naturally and under quinolone treatment, a melting temperature deviation from the expected melting temperature within the ±1°C range was still accepted as species-specific (methods subheading “Diagnostic accuracy estimation, agreement and comparison of obtained cycle threshold (Ct) values”, second paragraph, new fifth sentence).

Reviewer 1, fourth comment:

Delete line 265-281 as it is a repetition of line 248-264.

Authors:

The erroneously doubled text block has been removed.

Reviewer 1, fifth comment:

Line 341-343: add despite the long storage of extracted DNA

Authors:

As requested, the term has been added at the respective side.

Reviewer 1, sixth comment:

Line 372: delete “r” before “high”

Authors:

The typing error has been corrected as requested.

Reviewer 2 Report

Brief summary

To establish a diagnostic method for Arcobacter butzleri, the authors investigated several target genes for identification of the species. PCR targeting rpoB/C gave useful results, but the accuracy of the test for identification of the species was unsatisfactory. To compensate for this, a PCR targeting gyrA, which has excellent specificity, was suggested.

General comment

The background description is very well written and includes a comprehensive review of the Arcobacter butzleri. The description of methods and results is very clear, and the limitation statement is accurate and honest.

Unfortunately, the detection of target genes with both sensitivity and specificity, such that a diagnosis could be made in a single study, was not possible, but I felt that the paper was very well written as a scientific paper.

Specific comments

L241-245

The description at the top of the table is redundant.. Since the abbreviations are listed in the legend, it would be better to use abbreviations to make the table a little easier to read. Or, if the tables are swapped horizontally and vertically, it might be a little easier to read the tables.

Author Response

Reviewer 2, first comment:

L241-245. The description at the top of the table is redundant.. Since the abbreviations are listed in the legend, it would be better to use abbreviations to make the table a little easier to read. Or, if the tables are swapped horizontally and vertically, it might be a little easier to read the tables.

Authors:

As suggested, abbreviations explained in the legend were not explained in the table again and the table has been swapped horizontally and vertically.

Reviewer 3 Report

Dear authors,

your manuscript refers to comparing three real time-PCR methods, detecting three different genes, to confirm the presence of A. butzleri in fecal samples. The paper is well written, the Introduction gives insight into the topic, Materials and methods allow repeating the mentioned methods, and the Results are later clarified through tables and Discussion.

You explained the lack of this research well at the end of the paper, especially the lack of funds to carry out sequencing or connect the results  with regard to the source of the feces sample, which would give this research even more weight and significance.

Specific comments:

L 50-54-sentence too long

L 57 - less likely with less closely?

L 72 - dot missing at the end of the sentence.

L 73-nomenclature development?

L83- virulence factors with assumed relevance?

L 119, 121- twice accordingly

L 256 - mere?

-

Author Response

Reviewer 3, first comment:

L 50-54-sentence too long

Authors:

As requested, the sentence has been split.

Reviewer 3, second comment:

L 57 - less likely with less closely?

Authors:

As suggested, “less closely” was replaced “phylogenetically more distant”.

Reviewer 3, third comment:

L 72 - dot missing at the end of the sentence.

Authors:

The forgotten dot has been added.

Reviewer 3, fourth comment:

L 73-nomenclature development?

Authors:

This non-idiomatic and thus potentially confusing term has been removed.

Reviewer 3, fifth comment:

L83- virulence factors with assumed relevance?

Authors:

The non-idiomatic term “assumed relevance” has been replaced by “likely relevance”.

Reviewer 3, sixth comment:

L 119, 121- twice accordingly

Authors:

The second “accordingly” has been replaced by “in line with this”.

Reviewer 3, seventh comment:

L 256 - mere?

Authors:

The typing error has been corrected, “were” was meant.